[Supplementary Material]

# Learning to Poke by Poking: Experiential Learning of Intuitive Physics
# Supplementary Materials

**Pulkit Agrawal**[*]     **Ashvin Nair**[*]     **Pieter Abbeel**     **Jitendra Malik**     **Sergey Levine**
Berkeley Artificial Intelligence Research Laboratory (BAIR)
University of California Berkeley
{pulkitag,anair17,pabbeel,malik,svlevine}@berkeley.edu

The supplementary materials can be found at: http://ashvin.me/pokebot-website/.

---

[*]equal contribution, authors are listed in alphabetical order.