[Reviews · NeurIPS 2016]

Reviewer 1

Summary

The authors propose learning a forward model and an inverse model simultaneously to prevent collapse of the forward model and regularize the inverse model. They test this in the setting of a robot that can push objects; the task is to push the object from a given position to a target position.

Qualitative Assessment

This is a very nice, simple idea well presented. I have not seen it before, and think it will be useful to many others, as avoiding collapse of (deep) forward models is a standard annoyance. I don't really know anything about robotics, and don't feel competent to judge the difficulty of the problem they applied it to or the heuristic baseline they used, but the experiments seem compelling to me. It would be nice to see the idea evaluated in more environments or in other problems, and in particular, to get a better idea of how it breaks down as the space of actions becomes more complicated, but even as it is, I think this paper should be accepted.

Confidence in this Review

1-Less confident (might not have understood significant parts)


Reviewer 2

Summary

This paper presents an approach on how to learn to arange four objects by poking them with a robotic arm from visual input alone. The authors's model-based method jointly learns the forward and inverse dynamics. Instead of reconstructing images, the forward model predicts the feature vector of a layer in the deep neural network, that is supposedly more abstract due to the regularizing nature of the inverse model. They evaluate the policies defined by the trained inverse models on a simulation and on a lot of real-world data by measuring relative location error obtained through template matching. The paper concludes with a short section about active data collection instead of random sampling.

Qualitative Assessment

Pros: + A good idea to move away from generative models when they're not necesary + Real-world experiment with lots of data provides credible proof of concept Cons: - 'Strong' baseline is undefined. It's hard to judge the performance especially given the variances. - Unclear evaluation metric, because location and position terms seem to be mixed (l. 211, 212, 215 and Fig. 5 caption) and the influence of rotation is not clear. How exactly are the distances between object locations defined (d_ft and d_it) in terms of position and angle? - Using the median might indicate outliers that are not being discussed. Questions / Suggestions: - Separated evaluation in terms of position and angle might be useful. The shown trajectories seem to get the position right, but the angle is in some cases quite wrong. It would also be interesting to see if the angle is present in the representation x_t by comparing the activations while rotating the objects for the same position. - The claim in line 114 that, given x and u, a forward model is unique, only holds for single-agent Markov environments. What happens if the robot's arm is occluding the object? Minor issues: - The 'intuitive' part of intuitive physics is not clearly explained in the introduction. - Lots of language errors. Random or additional 'the' in many sentences (132, 208, 226, Fig4 caption). - The three streams in Figure 3 are not easily identifiable. - The oracle evaluation is explained in line 264, after everything has already been discussed. It might make sense to do this beforehand. - Reference [10] is Sascha Lange, as in [9].

Confidence in this Review

2-Confident (read it all; understood it all reasonably well)


Reviewer 3

Summary

This paper presents a deep neural network architecture that learns a combined forward and inverse model of object manipulation by "pokes". Pokes are actions in which a Baxter robot pushes an object with a stick attached to one of its end-effectors. The authors show that their siamese CNN based architecture allows the robot to position objects using a simple control policy. The system is trained on 50K random pokes on the physical platform.

Qualitative Assessment

This work is a nice demonstration of how recent advances in neural networks allow non-trivial robotics problems to be solved in a fairly standardized way. I really appreciate the non-trivial effort to evaluate the method on a real robotic system and the network architecture in Fig. 3 is quite clever. I do not agree that this system is learning "an internal model of intuitive physics". In the current setup, it's learning relationships between pairs of images. In my opinion "an internal physics model" implies that the system would learn some kind of representation of the forward and inverse dynamics of the robot and the objects being manipulated. Questions/remarks: - l 167: Why did you set lambda to 0.1. Did you optimize this parameter based on your simulation experiment? - You mention that your approach circumvents the hard problem of trajectory planning (l.138). However, the policy that you use during evaluation is in a sense a greedy trajectory planner (only looking one step ahead). The current policy is not guaranteed to guide the object to the target (e.g. if the table were to contain an obstacle). Therefore, it would be more correct to state that you do not consider the problem of trajectory planning here. - Fig. 4b seems to indicate that the forward model just speeds up learning. Would this also explain the difference between the inverse model and joint model results in Fig 5b? - Please fix the the annoying minor grammar issues ("its" vs "it's", "cmS" ).

Confidence in this Review

2-Confident (read it all; understood it all reasonably well)


Reviewer 4

Summary

The article proposes a very simple idea which is to jointly learn a forward model able to predict the next state of a system (given the action and the current state) and an inverse model able to predict which action has been applied between two states. This is justified by the fact that the author aims at learning directly from RAW inputs (i.e pixels) where it is very difficult to learn a forward model. Given this idea, and based on a classical convolutional architecture, the article evaluates this strategy onto a toy task and a real task which consists in learning to poke i.e to move objects on a table using a real robot. The experimental results clearly show that the strategy adopted here is relevant.

Qualitative Assessment

Comments: Even if the idea is very simple, I think it is a very interesting idea which can clearly contribute to the field. The paper is well written and gives strong arguments in favor of this research direction. I don’t see many comments to make on the 4 first sections of the paper and I must admit that I really liked to read this article. My only (but still important) concern is on the experimental part and on the way the resulting model is used. While the model is only learned given pairs of s_t and s_{t+1} during training (and predicts action a_t), it is then used to predict the best action allowing to move from a state to a particular goal, and the model is thus evaluated through it’s ability to solve long-term problems while it has been trained only on pairs of contiguous states. I am not sure if this experimental setup has consequences over the conclusion that using both inverse+forward models is interesting, but the article would be clearly stronger if the inverse model is learned based on a reinforcement learning setting (which is possible on the toy experiments, but much harder on the experiments with the robot). For example, and experimental protocol like the one in “Universal Value Function Approximators” (which basically proposes to learn a kind of inverse model based on a state and a goal) could be interesting. The way the evaluation is made here does not really allow us to draw general conclusions on the quality of the approach in the general RL case, but the approach is relevant for learning to poke, and certainly for similar robotics problems.

Confidence in this Review

3-Expert (read the paper in detail, know the area, quite certain of my opinion)


Reviewer 5

Summary

This paper proposed to predict actions (transformations) given the current and future states, and also to predict future states (in an embedded space) given the current state and the action. Both the inverse and the forward models are implemented as neural networks. Experiments on synthetic and real data shows the models help the robot to poke, performing better than a baseline.

Qualitative Assessment

This paper presents a very interesting idea on modeling the action space. I think the idea of learning actions from current and future states are novel, and the proposed realization is reasonable and intuitive. However, current experiments are limited and not convincingly demonstrate the advantage of the proposed framework. Specifically, the current framework looks like an intuitive and reasonable implementation of the idea. A possible flaw is the inferred action does not share the space with the ground truth action. It seems it's more natural (and may comes with better performance) if the model uses the inferred action and the current state to predict the future state, while posing losses on both the predicted action and the predicted state. The evaluations are unfortunately not very convincing. First, it seems the experiments are only on two objects under very constrained circumstances, which does not demonstrate the generalization power of the model. Second, among these two, the learned model (joint or inverse) does not outperform baseline significantly (the difference is indistinguishable for the hammer case, in specific). It is therefore unsure whether the model is learning significant informative signals. There is also no analysis or visualization of the learned representations. This paper is reasonably well written. The figures are illustrative, and the reference is thorough. Some additional reference is worth citing (Actions ~ Transformations from Wang and Gupta, CVPR16). I feel the current version is on the border, leaning slightly toward a poster but not against a rejection. This paper could be augmented by developing a more principled model, possibly coming with better performance, and more extensive experiments.

Confidence in this Review

3-Expert (read the paper in detail, know the area, quite certain of my opinion)


Reviewer 6

Summary

The authors propose a way to co-embed actions and state representations in order to improve a robotic manipulation task. The source of data is filmed unsupervised robot-object interaction. The novelty of the approach is using a combination of forward and inverse dynamics models on triplets of (poke, before image, after image) that creates useful visual representations during the action prediction stage and then learns a forward model in order to be able to predict the visual representations of the future states in order to be able to do inference. The experimental setup and methodology are sound and relevant and show that proposed approach leads to improvements over an inverse-only model and a strong hand engineered baseline.

Qualitative Assessment

I would spend more time on describing the inverse only model and why the authors think the additional forward model constraint leads to improvements. I would also spend more time on describing the human baseline and how it was generated.

Confidence in this Review

2-Confident (read it all; understood it all reasonably well)